# Connecting the Dots between Barriers to W.I.C. Access and Adult and Child Food Insecurity: A Survey of Missouri Residents

**DOI:** 10.3390/nu15112496

**Published:** 2023-05-27

**Authors:** Tyler L. Frank, Jason Jabbari, Stephen Roll, Dan Ferris, Takeshi Terada, Amanda Gilbert, Laura McDermott

**Affiliations:** 1Social Policy Institute, Brown School, Washington University in St. Louis, 1 Brookings Drive, St. Louis, MO 63130, USA; tylerfrank@wustl.edu (T.L.F.); stephen.roll@wustl.edu (S.R.); dan.ferris@wustl.edu (D.F.); terada@wustl.edu (T.T.); a.s.gilbert@wustl.edu (A.G.); 2Mathematica, P.O. Box 2393, Princeton, NJ 08543, USA; lmcdermott@mathematica-mpr.com

**Keywords:** food insecurity, W.I.C., maternal and child health, social determinants of health, program barriers

## Abstract

*Background.* Previous research has explored the impact of W.I.C. on recipients’ health, but less is known about the connection between barriers to W.I.C. access and health outcomes. We fill in a gap in the literature by studying the relationship between barriers to Special Supplemental Nutrition Program for Women, Infants, and Children (W.I.C.) access and adult and child food insecurity. *Methods.* After survey administration, we analyzed a cross-sectional sample of 2244 residents in Missouri who have used W.I.C. or lived in a household with a W.I.C. recipient in the past three years. We ran logistic regression models to understand the relationships among barriers to W.I.C. utilization, adult food insecurity, and child food insecurity. *Results.* Having special dietary needs (for adults), lacking access to technology, encountering inconvenient clinic hours of operation, and experiencing difficulties taking off work were associated with increased adult food insecurity. Difficulties finding WIC-approved items in the store, technological barriers, inconvenient clinic hours, difficulties taking off work, and finding childcare were associated with increased child food insecurity. *Conclusion.* Barriers to accessing and utilizing W.I.C. are associated with adult and child food insecurity. However, current policies suggest promising approaches to curbing these barriers.

## 1. Introduction

In the U.S., the Special Supplemental Nutrition Program for Women, Infants, and Children (W.I.C.) aims to improve the health of women, infants, and children younger than five years in lower-income households to support their unique nutritional needs [1]. Research shows that the W.I.C. program positively impacts health outcomes, including a lower risk of infant mortality, a lower risk of obesity, and reduced food insecurity [2,3,4]. Yet, despite these benefits, in 2019, less than 60% of eligible people participated in W.I.C. due to various barriers, including an inability to take time off from work, transportation problems, and misunderstanding the program [2,3,4]. While research has identified these barriers, relatively little work exists linking these barriers to health and nutrition outcomes [5,6]. Drawing on a survey of W.I.C. participants in the U.S. state of Missouri, we address this gap by descriptively examining the relationship between barriers to W.I.C. utilization and household food insecurity.

### 1.1. Background

The U.S. federal government established W.I.C. in the 20th century to address perceived deficiencies in the Food Stamp Program’s (now called SNAP) ability to address the unique needs of pregnant women and infants [7]. W.I.C. recipients must be either a pregnant person, a non-breastfeeding person up to six months postpartum, a breastfeeding person up to one year postpartum, an infant up to their first birthday, or a child up to their fifth birthday [7]. W.I.C. applicants must live within the state where they become eligible, document their household income, and recertify their eligibility every six or twelve months [7]. All W.I.C. state agencies set the income cutoff at no more than 185% of the federal poverty income guidelines [7]. The program offers free nutrition education, a supplemental food package, and referrals to health care and other services [7].

W.I.C. is one of the most extensive nutrition and food programs in the United States and serves more than 6 million people each month [5,8,9]. With its enormous reach, researchers have connected W.I.C. receipt with health improvements and food security for pregnant, postpartum, and breastfeeding individuals, infants, and children [8,10,11,12,13]. For example, studies have shown that W.I.C. is associated with a lower risk of food insecurity in pregnant and postpartum women, and the risk of postpartum household food insecurity was nearly 40% lower for women enrolling in W.I.C. during the first trimester compared to enrolling in W.I.C. during the third trimester [10,14].

At the same time, a substantial proportion of WIC-eligible individuals do not participate in the program. For example, less than 60% of eligible people in the U.S. participated in W.I.C., in part because of barriers including difficulty taking time off work to apply, lack of understanding of W.I.C., language and cultural barriers, transportation issues, the difficulty of the W.I.C. shopping process, and dietary issues [2,3,4]. Reducing barriers to W.I.C. enrollment may improve food security, increase food access, and reduce adverse maternal and child health outcomes [15,16,17,18,19]. However, there is limited research examining barriers to W.I.C. in detail and less evidence concerning the relationship between these barriers and health outcomes [19].

### 1.2. Theoretical Framework

Though food insecurity is a critical health issue for the population in general, the risks of food insecurity are compounded for pregnant women and young children as it increases the risk of adverse birth outcomes, such as infant mortality [20], and can harm early child development outcomes [21]. Maternal and child health outcomes in the U.S. are complexly related to the environmental, social, and material conditions in which families live and work—collectively known as the social determinants of health [22]. These social determinants are critical drivers of health disparities observed in the U.S., pronounced for women of color and Indigenous women, women of low socioeconomic status, and rural women [22]. Research has demonstrated that the social determinants of health are influenced by place and are connected to maternal and infant health outcomes [22]. For example, evidence shows that pregnancy-related hypertension mediated the relationship between racial residential segregation and infants’ low birthweight in Black women [22]. Additionally, low-income and Black residents who live in neighborhoods with concentrated deprivation have an increased risk of pregnancy-associated death [22].

The Geography of Opportunity is a theoretical framework emphasizing the importance of “place” in understanding health patterns, social determinants, and economic mobility. Recently, health researchers have adopted the Geography of Opportunity to make sense of unequal opportunity and lack of food options in neighborhoods. Acevedo-Garcia and colleagues define an opportunity neighborhood as “neighborhoods that support healthy development” [23]. A limited number of studies have used the Geography of Opportunity framework to understand the nuances of food insecurity [24,25,26,27]. We believe that the Geography of Opportunity offers a lens on the connection between food insecurity and barriers in accessing and utilizing W.I.C. As a result, we apply a Geography of Opportunity framework in two ways. First, we explore barriers to W.I.C. at multiple junctures along the W.I.C. journey, including barriers to W.I.C. clinics and barriers to W.I.C. retailers—each of which has a core relationship to place. In doing so, we explore barriers from a program lens, such as W.I.C. clinic location, and from an environmental lens, such as transportation. Second, we account for specific geographic indicators, such as urbanicity, in exploring W.I.C. barriers.

### 1.3. Current Study

We fill the research gap on W.I.C. barriers and health outcomes by surveying nearly three thousand residents in Missouri who have used W.I.C.—or live in a household with someone who has used W.I.C.—in the last three years. This survey asked participants about their experience with W.I.C., the extent to which they faced different barriers in accessing the program, and their experience with food insecurity. Using a descriptive research approach, we examine two research questions:To what extent do barriers to W.I.C. access relate to adult food insecurity?To what extent do barriers to W.I.C. access relate to child food insecurity?

## 2. Materials and Methods

### 2.1. Dataset

Data for this study come from the W.I.C. Experience Survey, a cross-sectional survey of residents in Missouri who (1) currently use W.I.C. or have used W.I.C. in the last three years; (2) reside in a household with someone who currently uses W.I.C. or has used W.I.C. in the last three years; or (3) are currently eligible for W.I.C. but are not participating in the program. Operation Food Search, a local nonprofit hunger relief organization, administered the online survey between 28 April 2022 and 21 June 2022. Operation Food Search recruited participants through three sources: (1) the W.I.C. Shopper App, a third-party mobile app that many state W.I.C. agencies promote, which helps W.I.C. recipients manage their benefits; (2) nonprofit organizations that serve low-income families; and (3) the Missouri W.I.C. Facebook page. Since participants were recruited from the W.I.C. Shopper App, nonprofit organizations, and the Missouri W.I.C. Facebook page, we could not calculate the estimates for response rates. Two thousand nine hundred fifty-six participants from Missouri completed the survey, and each participant was allowed to enter a drawing to win a USD 25 gift card. After listwise deletion of missing responses, 2244 respondents remained in the sample.

### 2.2. Measures

#### 2.2.1. Food Insecurity

The two outcome variables in this study were *adult food insecurity* and *child food insecurity*. Both variables come from the 18-item U.S. Household Food Security Survey Module [28]. Food security questions are ordered to group 10 adult-referenced questions before eight child-referenced questions [29]. This food security module asks about various food and nutrition experiences, including having money to buy food, affording balanced meals, cutting or skipping meals, losing weight due to hunger, and not being able to afford to feed children [29].

Respondents are then scored on their responses to the food security questions and categorized into one of four food security levels:

-High food security: no problems or anxiety about consistently accessing adequate food;

-Marginal food security: problems at times or anxiety about accessing adequate food, but the quality, variety, and quantity of their food intake are not substantially reduced;

-Low food security: reductions in the quality, variety, and desirability of diets, but the quantity of food intake and normal eating patterns are not substantially disrupted;

-Very low food security: the eating patterns of one or more household members have been disrupted, and food intake has been reduced because the household lacks money and other resources for food [30].

For our analysis, we describe households with high or marginal food security as “food secure” and those with low or very low food security as “food insecure” [30]. We coded food-secure households as “0” and food-insecure households as “1.”

#### 2.2.2. Barriers to W.I.C. Utilization

The survey asked participants multiple questions about barriers to W.I.C. use. We asked participants, “Have you been able to find all the WIC-approved items you want to buy at one grocery store?” Responses were “Yes,” “No,” and “Not sure.” Moreover, participants were asked, “Do you or your child have any special dietary needs? Please select all that apply.” The options were “Yes, myself,” “Yes, my child,” and “No.” These different responses were recoded as separate dichotomous variables; i.e., the adult has dietary needs (Yes or No), and the child has dietary needs (Yes or No). Additionally, we asked participants, “Have any of the following been barriers to you being able to access a W.I.C. clinic?” Barriers included access to technology (e.g., for locating clinics or making appointments), access to transportation, clinic locations, clinic hours of operation, clinic wait times, having to take off time from work, and having to find childcare. Options were “No,” “Yes,” and “Not sure.” We combined the “No” and “Not sure” responses into the reference category (coded as 0); “Yes” was coded as 1.

#### 2.2.3. Household Characteristics

The survey asked respondents several demographic questions connected with social determinants of health and related to the Geography of Opportunity. These questions documented participants’ race/ethnicity, whether they lived in a single-parent household, pre-tax income, employment situation (e.g., full-time, part-time, and unemployed), and the number of adults and children in the home. Participants could also indicate who in the household was using W.I.C.: a pregnant person, a postpartum person (up to six months after the end of pregnancy), a breastfeeding person, a baby (up to their first birthday), or a child (up to their fifth birthday). We coded each option as a binary Yes/No indicator. We used rural–urban commuting area codes (RUCAs) to measure urbanicity—accounting for a U.S. Census tract’s population density, urbanization, and commuting [31]—to classify participants’ zip codes as metropolitan or non-metropolitan.

Finally, we asked participants about their participation in a range of government benefit programs, including the Supplemental Nutrition Assistance Program (SNAP), Temporary Assistance for Needy Families, Unemployment Insurance, Public Housing or Housing Choice Voucher (e.g., Section 8), LIHEAP/Utility Assistance Program, Child Care Subsidy Program in Missouri, Head Start, VA Benefits, Medicare, Medicaid, Social Security (e.g., supplemental, survivor, retirement, disability), and National School Lunch and Breakfast Programs. Responses were coded as “Yes” and “No” for each separate government program. Then, we created a continuous variable, which counted the number of programs respondents participated in.

### 2.3. Analytic Approach

We ran logistic regression models in STATA for *adult* and *child food insecurity* outcomes. Equation (1) displays the format of the logistic regression models:(1)$$log\left(\frac{P}{1-P}\right)=\beta_{0}+{\gamma x}_{i}^{WICbarriers}+\pi x_{i}^{Covariates}$$

The equation shows that the probability of *adult food insecurity* or *child food insecurity* for a respondent, *i*; *Pr*(*Y_i_ =* 1), is a function of barriers to W.I.C. utilization and the set of sociodemographic covariates.

The sociodemographic covariates include single-parent status, race/ethnicity, income, government benefits, number of adults and children in the household, employment status, the type of person using W.I.C. benefits, and urbanicity. We chose predictors from questions in the survey that may correlate with barriers to W.I.C. enrollment and our food insecurity outcomes.

## 3. Results

Table 1 presents the descriptive statistics of respondents from the survey. In total, 25% of participants lived in a single-parent household. Additionally, 70% of participants were white, over 15% were Black, less than 10% were Hispanic or Latinx, 3% were Asian, and the remaining 3% were Middle Eastern, Northern African, Native Hawaiian, Pacific Islander, American Indian, Alaskan Native, or another race. More than 20% of households had a pregnant person, almost 30% were postpartum, more than 30% were breastfeeding, more than 25% had a baby up to the first birthday, and approximately 25% had a child up to the fifth birthday. Less than half of the sample had one or two adults in the household, and more than half had three or more adults in the home. Relatedly, over 60% of households had one to two children, and almost 40% had three or more children. Approximately 35% of households had a monthly income between USD 0 and 2000, over 30% had an income between USD 2001 and 4000, and more than 30% had an income of USD 4001 or higher. Nearly 40% of participants were employed full-time, almost 40% were employed part-time, and more than 20% were unemployed. Almost 70% of respondents lived in a metropolitan area. The mean number of government benefit programs for the sample was four, and the range of program receipt was zero to twelve.

For households that filled out the W.I.C. Experience Survey, nearly 60% experienced adult and child food insecurity in the last twelve months. Barriers to W.I.C. utilization were multifaceted. More than 20% could not find WIC-approved items in the store, 20% had adult special dietary needs, and 20% had child special dietary needs. Almost 40% experienced barriers in access to technology, nearly 40% experienced barriers in access to transportation, and 35% experienced barriers in clinic locations. In total, 40% of respondents experienced barriers in clinic hours of operation, nearly 40% experienced barriers in clinic wait times, almost 40% experienced barriers in having to take off time from work, and close to 30% experienced barriers in finding childcare.

Table 2 includes the results of the logistic regression models of *adult food insecurity* and *child food insecurity* outcomes. In the *adult food insecurity* model, adults with special dietary needs were associated with a 92% increase in the odds of experiencing adult food insecurity compared to adults without special dietary needs (*p* < 0.001). Participants with technology access barriers were associated with a 45% increase in the odds of having adult food insecurity compared to participants without this barrier (*p* < 0.01). Participants who reported clinic hours of operation as a barrier were associated with a 23% increase in the odds of experiencing adult food insecurity compared to other participants (*p* < 0.05). Moreover, respondents who found it difficult to take time off from work were associated with a 35% increase in the odds of having adult food insecurity compared to participants who did not (*p* < 0.01).

In the *child food insecurity* model, participants who did not find WIC-approved items at one grocery store were associated with a 29% decrease in the odds of experiencing child food insecurity compared to participants who found all WIC-approved items (*p* < 0.05). Participants with technology access barriers were associated with a 43% increase in the odds of having child food insecurity, compared to participants without that barrier (*p* < 0.01). Respondents who reported clinic hours of operation as a barrier were associated with a 28% increase in the odds of experiencing child food insecurity compared to participants who did not (*p* < 0.05). Participants who found it difficult to take time off from work were associated with a 46% increase in the odds of having child food insecurity compared to participants who did not (*p* < 0.01). Additionally, respondents reporting childcare barriers were associated with a 47% increase in the odds of experiencing child food insecurity compared to respondents who did not (*p* < 0.01).

## 4. Discussion

Although W.I.C. is associated with increased nutrition and decreased food insecurity, barriers prevent eligible families from receiving and utilizing this benefit [32,33]. However, research has yet to demonstrate how these barriers affect health outcomes. We fill the gap by exploring how W.I.C. barriers relate to food insecurity in a statewide survey of WIC-eligible households. Through our research partnership with a regional hunger-relief organization and the Missouri Bureau of W.I.C. and Nutrition services, our findings can directly inform state-level policies and practices related to W.I.C. administration. Our results also provide critical insights into removing barriers to W.I.C. for policymakers, stakeholders, other states, and federal departments, such as the U.S. Department of Agriculture.

Within individual barriers, our findings demonstrate that a lack of technological access was associated with increased food insecurity. Technology issues may reflect the inability of participants to locate essential information related to W.I.C. (e.g., clinic and retailer locations). Moving on to W.I.C. clinics, our findings demonstrate that limited hours of operation and having to take time off from work and find childcare were associated with increased food insecurity. These barriers are likely related, as expanded hours of operation may reduce the necessity of taking off work and finding childcare. Concerning W.I.C. retailers, our findings demonstrate that adults having special dietary needs were also associated with increased adult food insecurity, which could reflect limited food options for these individuals. We were surprised that not finding WIC-approved items was associated with decreased food insecurity for children. However, it could be the case that more frequent or deliberate W.I.C. users—who may have an easier time finding approved items—have additional needs related to increased food insecurity.

In light of these findings, there are several implications for policy and practice. First, given the importance of technology and difficulties in taking off work and finding childcare, state agencies should consider leveraging web-based solutions to make accessing and utilizing W.I.C. easier. For example, recent research has demonstrated that individuals that use an online W.I.C. shopping mobile application are more likely to redeem full benefits [34]. Of course, as not everyone has access to technological tools or high-speed internet, broader efforts to expand technology—as a core social determinant of health—should be considered [35]. A recent USDA grant of USD 502 million to provide high-speed internet in rural communities across 20 states represents a recent example of these efforts [35]. Moreover, policymakers should consider ways to increase opportunities for online ordering [36]. Online ordering may also help reduce barriers for participants with special dietary needs or those who otherwise face difficulties finding WIC-approved items in their local retailer.

Furthermore, W.I.C. clinics should consider efforts to increase hours for families. While few studies have explored expanded clinic hours, efforts to reduce wait times were recently explored. Using statistical process control techniques, Boe and colleagues reduced wait times in a county health department W.I.C. clinic by four minutes [37]. As wait times are associated with overall W.I.C. satisfaction [37], reducing wait times may increase W.I.C. participation. Given the importance of childcare in accessing W.I.C. [38], it is surprising that we could not find any examples of on-site options. Nevertheless, given the effectiveness of on-site childcare options in employment [39] and other work-related settings—including academic health centers [40]—W.I.C. agencies should consider implementing on-site childcare options.

Additionally, our findings have implications for W.I.C. retailers. For example, recent research has explored the barriers and subsequent opportunities for WIC-approved item placement [6]. Here, retailers may consider increasing the use of “Shelf Talkers” (special tags that designate WIC-eligible items), creating a special section that only contains WIC-approved items, and placing WIC-approved items closer to the register [6]. Finally, while W.I.C. packages have been updated over the years to meet special dietary needs [41], continued efforts to meet beneficiaries’ preferences should be considered [42].

Despite this study’s implications, it is not without limitations. Concerning internal validity, while we account for various sociodemographic characteristics related to W.I.C. barriers and food insecurity in our models, we cannot establish causality. Instead, our findings highlight descriptive relationships. Concerning external validity, despite our large and broad sample, our focus on a single state can limit the generalizability of our results.

Given the relationship between barriers to W.I.C. and food insecurity, future research should continue exploring both barriers to obtaining and utilizing W.I.C. and opportunities to remove these barriers. Some of these opportunities have occurred in recent innovations during the COVID-19 pandemic. While the COVID-19 pandemic has posed a severe health risk, it has shed light on the importance of public benefit programs while spurring novel technological innovations. For example, Jabbari and colleagues found that increased fruit and vegetable vouchers were associated with improved nutrition [32].

Furthermore, when states offered remote W.I.C. appointments during the COVID-19 pandemic, participants reported increased satisfaction and reduced compliance costs for accessing W.I.C. benefits [43]. Hybrid and remote approaches to accessing W.I.C. benefits during the pandemic were also associated with increased certification and education completion rates [44]. As a review of W.I.C. interventions confirmed the importance of virtual strategies in accessing and utilizing W.I.C. benefits [15], these strategies should not only be considered a stop-gap for COVID-19 but a long-term strategy for improving child and maternal health. Thus, while our study is one of the first to demonstrate the relationship between barriers to W.I.C. and food insecurity, it is only the first step to improving health outcomes for low-income families. Given that nearly 60% of our sample experienced food insecurity and, in many cases, almost 40% experienced a barrier to utilizing W.I.C., future research should continue exploring novel approaches to remove W.I.C. barriers to improve health outcomes for women, infants, and children.

## 5. Conclusions

Previous research has explored the health impacts of W.I.C., but less is known about the connection between barriers to W.I.C. access and health outcomes. In a statewide survey of Missouri residents, we fill this gap through a descriptive study of barriers to W.I.C. access and their relationship to household food insecurity. Barriers to finding WIC-approved items in the store, barriers to adult dietary needs, technological barriers, inconvenient clinic hours of operation, difficulties taking off work, and issues in finding childcare were associated with adult and child food insecurity. These results reflect barriers in W.I.C. users, clinics, and retailers and have implications for practice and policy. To increase the utilization of W.I.C. in Missouri, W.I.C. agencies and policymakers must be aware of and reduce barriers that deter eligible recipients from participating.

## Figures and Tables

**Table 1 nutrients-15-02496-t001:** Baseline characteristics of participants from the W.I.C. Experience Survey.

Descriptive Statistics	N	%
		(Mean, Range)
Adult Food Security		
Low or very low	1320	58.82
High or marginal	924	41.18
Child Food Security		
Low or very low	1315	58.60
High or marginal	929	41.40
Barriers to W.I.C. Utilization		
Cannot find WIC-approved items	475	21.17
Adult has special dietary needs	459	20.45
Child has special dietary needs	463	20.63
Lacks access to technology	839	37.39
Lacks access to transportation	844	37.61
Inconvenient clinic locations	776	34.58
Inconvenient clinic hours of operation	899	40.06
Long clinic wait times	849	37.83
Difficulties taking off work	869	38.73
Difficulties finding childcare	748	33.33
Single-parent household (Yes)	563	25.09
Race and Ethnicity		
White or Caucasian	1560	69.52
Black or African American	355	15.82
Hispanic or Latinx	178	7.93
Asian	73	3.25
Other race	78	3.48
Person using W.I.C. (Yes)		
Pregnant person	521	23.22
Postpartum person (up to 6 months)	648	28.88
Breastfeeding person	720	32.09
A baby (up to the first birthday)	607	27.05
A child (up to the fifth birthday)	584	26.02
Adults in Household		
One	286	12.75
Two	731	32.58
Three	436	19.43
Four or more	791	35.25
Children in Household		
One to two	1362	60.70
Three or more	882	39.30
Income		
USD 0–1000	323	14.39
USD 1001–2000	485	21.61
USD 2001–3000	396	17.65
USD 3001–4000	311	13.86
USD 4001–5000	322	14.35
USD 5001 or higher	407	18.14
Employment		
Full-time	865	38.55
Part-time	867	38.64
Unemployed	512	22.82
Urbanicity		
Metropolitan	1565	69.74
Non-metropolitan	679	30.26
Government Benefit Programs	2244	(3.98, 0–12)

*Note.* N = 2244.

**Table 2 nutrients-15-02496-t002:** Logistic Regression Models for Adult and Child Food Insecurity Outcomes.

Effect	Adult Food Insecurity		Child Food Insecurity	
	OR (SE)	*p*	OR (SE)	*p*
Barriers to W.I.C. Utilization (reference = No)				
Cannot find WIC-approved items	1.160 (0.138)	0.212	0.714 (0.094)	0.010
Adult has special dietary needs	1.915 (0.243)	0.000	1.178 (0.162)	0.234
Child has special dietary needs	1.069 (0.125)	0.568	0.946 (0.122)	0.664
Lacks access to technology	1.445 (0.156)	0.001	1.433 (0.190)	0.007
Lacks access to transportation	1.114 (0.114)	0.294	1.216 (0.151)	0.115
Inconvenient clinic locations	0.970 (0.105)	0.780	1.181 (0.156)	0.208
Inconvenient clinic hours of operation	1.232 (0.122)	0.035	1.282 (0.154)	0.039
Long clinic wait times	1.113 (0.113)	0.295	1.179 (0.147)	0.186
Difficulties taking off work	1.348 (0.134)	0.003	1.461 (0.173)	0.001
Difficulties finding childcare	1.072 (0.110)	0.497	1.467 (0.183)	0.002
Household Characteristics	Yes		Yes	
Intercept	1.076 (0.250)	0.754	0.202 (0.055)	0.000

*Note.* Household characteristics include the type of person using W.I.C., the number of adults and children in the household, income, employment status, government benefit programs, single-parent household, and race and ethnicity. N = 2244.

## Data Availability

The data presented in this study are available on request from the corresponding author.

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
