# Peer review of "Connecting the Dots between Barriers to W.I.C. Access and Adult and Child Food Insecurity: A Survey of Missouri Residents"

_nutrients, 2023, doi:10.3390/nu15112496_

Round 1

Reviewer 1 Report

The authors conducted an important study to determine the relationship between WIC access and adult/child food insecurity. A well sampled, large number of participants were surveyed to identify the impact of barriers to access and key health outcomes. Overall, I found the paper to be very well written, scientifically conducted with appropriate rigor, and with an appropriate interpretation of the results, which provided clear actionable direction for both future research and practical considerations to address the problems being analyzed. 

The analytical approach was well described with appropriate statistical testing and clarity to fully understand the results. Tables and figures were well constructed. A minor comment is the formatting for CI was off and that may have occurred during uploading of documents. No additional concerns. 

Author Response

Thanks for your feedback. The tables have been updated to avoid confusion in the CI.

Warmly,

Tyler

Reviewer 2 Report

The paper is interesting, well written and sound. In many parts, the authors are writing for US readers. Nutrients is a high impact journal for a large audience worldwide. I commend the authors for improving their writing for a broader audience.

I have made some suggestions; I hope you find them helpful:

Title: WIC is not common jargon worldwide. I suggest using "Supplemental Nutrition Program" or similar to be understandable to a wider audience.

Something is wrong with the affiliations. Only the address is shown. What college or institute are the authors from?

Abstract: Please provide the WIC definition

Please provide more information about the methods. Did you use questionnaires? Face to face? What did you ask about?

Introduction - Please state that WIC is a program from the US.

I think you could provide some information about what WIC is. You describe the barriers, but I do not know the program (since I am not a US citizen). It is difficult to understand.

First sentence of background - Add that it is the federal government of the US.

The methods are very clear!

I had difficulty understanding tables 2 and 3. They seem skewed (especially 95% CI). If you reduce the spacing, the table could fit on one page.

Personally, I find the use of both 95% CI and p-value superfluous for reporting OR. Is not just 95%CI sufficient?

Congratulations it is a very interesting study.

Author Response

Thank you for your feedback. The manuscript has been updated to include more inclusive definitions and terminology for an international audience. Also, the author affiliations have been updated, as well as expanded methods, WIC definition, and revised tables to improve clarity in methods and results.

Warmly,

Tyler